# Computational Strategies for Assessing Adverse Outcome Pathways: Hepatic Steatosis as a Case Study

**DOI:** 10.3390/ijms252011154

**Published:** 2024-10-17

**Authors:** Rita Ortega-Vallbona, Martina Palomino-Schätzlein, Laia Tolosa, Emilio Benfenati, Gerhard F. Ecker, Rafael Gozalbes, Eva Serrano-Candelas

**Affiliations:** 1ProtoQSAR S.L., Calle Nicolás Copérnico 6, Parque Tecnológico de Valencia, 46980 Paterna, Spain; rortega@protoqsar.com (R.O.-V.); mpalomino@protoqsar.com (M.P.-S.); rgozalbes@protoqsar.com (R.G.); 2Unidad de Hepatología Experimental, Instituto de Investigación Sanitaria La Fe (IIS La Fe), Av. Fernando Abril Martorell 106, 46026 Valencia, Spain; laia_tolosa@iislafe.es; 3Biomedical Research Networking Center on Bioengineering, Biomaterials and Nanomedicine (CIBER-BBN), Instituto de Salud Carlos III, C/Monforte de Lemos, 28029 Madrid, Spain; 4Istituto di Ricerche Farmacologiche Mario Negri IRCCS, Via Mario Negri 2, 20156 Milan, Italy; emilio.benfenati@marionegri.it; 5Department of Pharmaceutical Sciences, University of Vienna, Josef-Holaubek Platz 2, 1090 Wien, Austria; gerhard.f.ecker@univie.ac.at; 6MolDrug AI Systems S.L., Olimpia Arozena Torres 45, 46108 Valencia, Spain

**Keywords:** adverse outcome pathway, hepatic steatosis, computational toxicity, new generation risk assessment, molecular mechanisms

## Abstract

The evolving landscape of chemical risk assessment is increasingly focused on developing tiered, mechanistically driven approaches that avoid the use of animal experiments. In this context, adverse outcome pathways have gained importance for evaluating various types of chemical-induced toxicity. Using hepatic steatosis as a case study, this review explores the use of diverse computational techniques, such as structure–activity relationship models, quantitative structure–activity relationship models, read-across methods, omics data analysis, and structure-based approaches to fill data gaps within adverse outcome pathway networks. Emphasizing the regulatory acceptance of each technique, we examine how these methodologies can be integrated to provide a comprehensive understanding of chemical toxicity. This review highlights the transformative impact of in silico techniques in toxicology, proposing guidelines for their application in evidence gathering for developing and filling data gaps in adverse outcome pathway networks. These guidelines can be applied to other cases, advancing the field of toxicological risk assessment.

## 1. Introduction

The evolving landscape of risk assessment is leading to the development of hypothesis-driven, tiered, and iterative approaches to prioritize chemicals. These new early toxicity detection strategies, incorporating New Approach Methodologies (NAMs), offer an ethical alternative to traditional in vivo experimentation, which faces challenges in ethics, time, and cost [1,2]. NAMs include tools such as in silico models, artificial intelligence, machine learning (ML) systems, toxicogenomic tests, and read-across approaches [1,3,4]. These human-relevant tools can transform safety assessments by offering insights into key biological pathways and targets [1,4,5]. NAMs mimic human biology and provide mechanistic information on chemical toxicity [1], which can be organized into Adverse Outcome Pathways (AOPs) and AOP networks, establishing relationships between the interaction of a chemical with a biological system and the resulting adverse outcome (AO) [5,6,7].

AOPs go beyond the older mode-of-action concept by considering effects at varying levels of biological organization [2,8,9]. A molecular initiating event (MIE) is always the first step of an AOP, where the chemical modulates with a specific molecule in the biological system [7]. The AO is the apical toxicological endpoint, referring to an acute or local adverse effect or systemic outcome. The pathway from an MIE to an AO consists of a series of steps involving different biological elements that can lead to adverse health effects. Some of those steps are considered key events (KEs), which are measurable changes in a biological state that are essential but not sufficient for the AO to occur [8,10,11] and are connected by KE relationships. A KE characterizes the system’s condition at a specific stage, whereas a KE relationship scientifically explains how a system transitions from one upstream KE to another downstream KE. It establishes a connection between the two events, facilitating the inference or extrapolation of the downstream KE’s state from the known, measured, or predicted state of the upstream KE [5,11]. In addition, a quantitative AOP involves computational models that quantitatively characterize the relationships between MIEs and KEs, as well as among KEs and the AO [11]. A quantitative AOP provides quantitative, dose-responsive, and time-course predictions [2,12,13].

An AOP is not a test method, computational model, or risk assessment but rather a structured description of collective knowledge in toxicology supported by scientific evidence. AOPs are designed to aid regulatory decision making using mechanistic data rather than solely outcome data from traditional whole-organism toxicity tests [1,8,14]. However, it is important to note that AOPs themselves do not include exposure evaluation, but they can be used in conjunction with exposure data to inform risk assessment processes [14]. AOPs facilitate the prioritization of chemicals for further testing through screening processes to determine if they trigger any biological response. The substances can be ranked according to their toxicological risk to guide detailed testing of higher-risk compounds. In the same way, AOPs can be used to label chemicals based on their potency to activate an AO [1,5].

To improve chemical safety assessment through AOPs, it is necessary to gather and integrate biological process knowledge into a cohesive network [1]. AOPs can be combined into networks that describe common KEs activated by different MIEs to provide a complete view of the mechanisms leading to pathology [10,11]. Developing an AOP requires collecting information on each building block from the literature, in vivo or in vitro assays, or computational methods and associating evidence and testing methods [7]. AOPs are “living documents” and should be continually refined as new data become available [11,15]. Despite ongoing efforts to link test methods and data with AOPs, current initiatives have limitations, relying on publicly available knowledge without sufficient curation for fidelity. These initiatives often overlook evidence from in silico predictive systems and struggle to associate diverse data formats from various NAMs [1].

Diverse computational methods can be employed to identify AOP components, predict the effects of chemicals on each node at the molecular level, and fill knowledge gaps within an AOP network. Figure 1 shows schematically which computational methods can be applied in each case.

Fatty liver disease or hepatic steatosis is defined as the accumulation of fatty acids (FA) in the liver beyond 5% of its wet weight due to an abnormality in the synthesis, transportation, elimination of lipids, or a combination of those factors [16]. Steatosis can be defined as increased fat accumulation in the liver without hepatocellular necrosis and no, or minimal, inflammation. Although it can be a mild condition with a good short-term prognosis, it can lead to steatohepatitis lesions characterized by hepatocellular injury and inflammation [17,18,19]. Liver steatosis exhibits two histological patterns: macrovesicular steatosis, marked by large vacuoles causing peripheral nucleus displacement, and microvesicular steatosis, characterized by numerous smaller vacuoles without nucleus displacement [19,20]. Several causes can lead to hepatic steatosis, such as obesity, insulin resistance, and exposure to alcohol and chemicals, including pharmaceuticals [19,21,22].

Drug-induced steatosis has the potential to directly cause metabolic-associated steatotic liver disease (MASLD, formerly known as non-alcoholic fatty liver disease). Even though drugs are not typically a primary cause of MASLD, it has been demonstrated that they might underlie, mimic, or exacerbate MASLD [23]. It is known that certain drugs commonly used in clinical practice, such as amiodarone, tamoxifen, or certain chemotherapeutic agents, may contribute to hepatic steatosis. Drug-induced steatosis is a well-studied condition, and the mechanisms behind it have been extensively described in various AOPs. These AOPs have been combined into a comprehensive AOP network, which is depicted in Figure 2 [11,24,25,26].

FA accumulation, which is a major feature of this condition, is caused by an increase in de novo FA synthesis and a decrease in β-oxidation, both of which indicate a disruption in FA metabolism, as well as an increase in FA influx [22,27].

The primary MIEs of this AOP are believed to result from the interaction between chemicals and various nuclear receptors. The activation of the liver X receptor (LXR), a transcriptional regulator of genes linked to cholesterol uptake, transport, and excretion, is an integral part of the AOP [20,28]. LXR activation upregulates four genes involved in de novo FA synthesis and increases hepatic FA uptake via the CD36 (Cluster of Differentiation 36, also known as Fatty Acid Translocase (FAT)) [19,20,22].

Other receptors have been described as causing the accumulation of triglycerides by sharing one or both mechanistic pathways described for LXR. These receptors include the aryl hydrocarbon receptor (AHR), the constitutive androstane receptor (CAR), the farnesoid X receptor (FXR), the glucocorticoid receptor (GR), the peroxisome proliferator-activated receptors (PPARs), and the Pregnane X receptor (PXR) [19,22]. CAR, FXR, LXR, PXR, and PPARs share some characteristics: they are nuclear receptors of the subfamily 1 and heterodimerize with retinoid X receptor (RXR). It has been reported that RXR competition could play a functional role in lipid metabolism [22,28,29].

Mitochondria play a pivotal role in maintaining lipid homeostasis, and disruptions in their function can lead to the accumulation of lipids within cells. Some hepatotoxicants induce mitochondrial dysfunction, inhibiting mitochondrial β-oxidation and resulting in the intracellular accumulation of FA. Activation of PPARα by FA and fibrate drugs regulates FA oxidation pathways, while antagonistic binding inhibits fatty acyl-CoA oxidase, suppressing microsomal β-oxidation and promoting triglyceride accumulation [19,22]. The estrogen receptor (ER) also regulates appetite and adipose biology, which has a close relationship with FA uptake by the hepatocytes [22].

The ASPIS cluster, created in the EU Horizon 2020 Research and Innovation programme, strives to advance the safety assessment of chemicals while minimizing the reliance on animal testing (https://aspis-cluster.eu/, accessed on 20 September 2024). ASPIS is a collaboration between the projects ONTOX (https://ontox-project.eu/, accessed on 20 September 2024), PrecisionTox (https://precisiontox.org/, accessed on 20 September 2024), and RISK-HUNT3R (https://www.risk-hunt3r.eu/, accessed on 20 September 2024). The ASPIS cluster creates a network of NAMs using case studies to predict various AOs, including drug-induced hepatic steatosis.

This paper reviews the use of computational techniques to fill in the missing data gaps in the AOP network for drug-induced hepatic steatosis. The results of this study provide valuable insights into the potential of these techniques to improve our understanding of AOPs and their role in risk assessment. This could have significant implications in the toxicology field by enabling more accurate predictions of chemical-induced toxicity and the development of targeted risk management strategies. We propose establishing guidelines for using in silico strategies to gather the weight of evidence, which can be applied to other cases where an AOP network has been developed.

## 2. Computational Methods

### 2.1. Automatized SAR

The structure–activity relationship (SAR) is a technique based on the connection between the chemical structure of a molecule and its physicochemical, biological, or toxicological activity. The main idea behind the SAR is that certain structural features determine significant changes in its biological effects. The molecular substructures responsible for the toxicological effects of a compound are considered “structural alerts” (SAs) [30,31]. SAs are typically selected based on scientific publications, expert knowledge, and understanding of structure–activity relationships, toxicological pathways, and metabolism [32,33]. Alternatively, various statistical pattern recognition techniques can automatically mine SAs from a dataset of molecules with clear structural information and associated bioactivity values [32,34].

For automatic SA generation (automatized SAR), substructures can be identified using different approaches. Table 1 shows a range of algorithms for the automatic generation of SAs that use different approaches, such as utilizing predetermined fragments, embedding molecules into graphs, or employing sets of fingerprints [30,35]. Some researchers have used strategies based on established fingerprints, like PubChem’s 881-bit fingerprint indicating unique substructures [36] and Klekota’s fingerprint encoding 4860 substructures [37].

Once the set of SAs has been established, statistical metrics such as precision, enrichment factor, and p-value are used to assess their performance [30]. Additionally, as is characteristic in computational models based on a chemical structure, SAR models have an applicability domain (AD) [45] that defines the chemical space in which the model’s predictions are reliable, and different methods can be applied to estimate it, including leveraging the range of descriptors, calculating leverage values, defining descriptors subspaces, or establishing confidence indices based on nearest neighbours. These methods have been extensively discussed elsewhere [46,47].

An essential advantage of SAR analysis lies in the rapid computational detection of these rules in a new set of molecules [33]. Additionally, SAR analysis enables mechanistic interpretation, associating SAs with specific biochemical mechanisms such as enzyme activation or ion channel opening [48,49]. This association helps to identify chemicals sharing a common mechanism of action [31]. The use of molecular fragments or fingerprints as SAs contributes to visualizing the SAR in terms of actual chemical structure. A simple 2D representation of a molecule with the relevant SAs highlighted (Figure 3) makes it easier for the user to understand complex chemical interactions. This facilitates the interpretation and enhances accessibility [50].

Nevertheless, SAs cannot give quantitative results; they can only classify compounds into discrete groups. To cope with this issue, QSARpy has been introduced, associating with the fragment a quantitative value related to the potency of the effect [51]. On top of that, they often give inconclusive results for negative predictions, as the absence of an SA in a compound is not enough proof to classify it as negative [17,31]. Therefore, they are a weak option when trying to generate data for constructing a quantitative AOP. Furthermore, expertise in cheminformatics and toxicology is required to create SAs and understand their underlying mechanisms [17], which can make developing and updating SAR models time-consuming [31].

**Figure 3 ijms-25-11154-f003:**
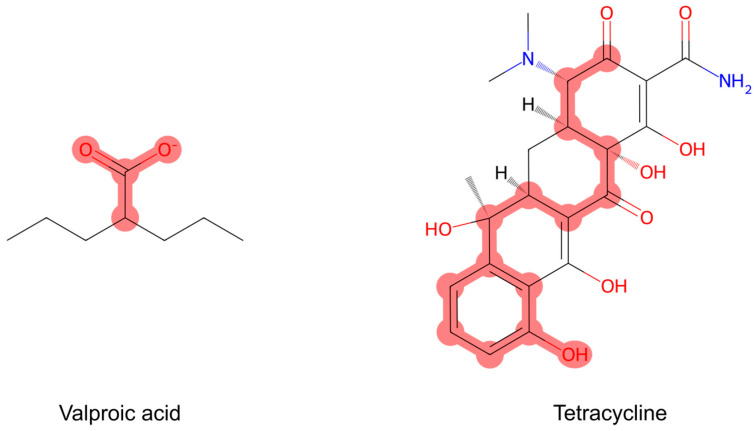
Representation of two steatotic compounds, valproic acid and tetracycline, showing two structural alerts for PPAR binding from Mellor et al., 2016 [52]. The carboxylic acid group (SMARTS: C(=O)O) is highlighted in valproic acid. A retinoid substructure is highlighted in tetracycline (SMARTS: [#6]~*~[#6]~*~[#6]~*~[#6]~*~a1aa([O,Cl,F,I,Br,N*])aaa1).

To illustrate the application of SAR analysis in understanding hepatic steatosis, this section explores examples where researchers have worked to relate SAs with different points of the steatosis AOP. Mellor and colleagues characterized 214 SAs associated with binding to 10 nuclear receptors recognized as MIEs in steatosis AOPs: AHR, CAR, ER, FXR, GR, LXR, PPAR, PXR, retinoic acid receptor (RAR), and RXR [52]. Information on agonists for these receptors was gathered from ChEMBL and PDB. First, chemical descriptors were calculated for the ligands to define the chemical space covered by them. Next, structural features associated with the protein–ligand interactions were identified and encoded into SMARTS strings, setting the group of SAs that would be applied to an in silico workflow. For most nuclear receptors, the substructural features generally follow a consistent pattern of examining key scaffolding structures (e.g., ring structures) and subsequent screening for essential functional groups. Exceptions include RXR and PPAR, which have specific exclusion rules, as well as GR, AR, ER, and PXR, which implement high molecular weight filters to accommodate larger ligands with distinct receptor binding patterns compared to those with lower molecular weights.

Although not focused solely on steatosis, a publication by Ivanov and colleagues demonstrated the use of PASS software (https://www.way2drug.com/, accessed on 20 September 2024) to identify protein targets related to drug-induced liver injury (DILI) [53]. They analyzed a set of 699 drugs classified into three DILI severity classes (no DILI, moderate, and severe). Using the PASS Targets version, they predicted interactions of these drugs with a total of 1534 human proteins. Among these, they detected 61 proteins associated with DILI. Notably, out of these 61 proteins, 10 were found to be related to lipid disorders, which involve a pathological mechanism closely linked to steatosis.

Some of the proteins related to the lipid disorders they identified are particularly interesting: AHR is an MIE for the steatosis AOP; fructose-1,6-biphosphatase takes part in the gluconeogenesis pathway, which, when it is excessive, can promote de novo lipid synthesis, leading to steatosis [54]; dihydroorotate dehydrogenase is involved in the de novo pyrimidine biosynthesis pathway and is also coupled to the mitochondrial respiratory chain [55]; some other proteins identified are related to MASLD—and not exclusively to drug-induced steatosis. This study is an excellent example of how SAR models are useful for identifying SAs and characterizing possible MIEs for steatosis.

Mitochondrial dysfunction is one of the KEs in the AOP network that has been most largely studied. In a book chapter published in 2018, Enoch, Mellor, and Nelms compiled a list of 22 SAs detected on different datasets of chemicals classified as active or inactive for mitochondrial toxicity [56]. According to their findings, these SAs primarily address two specific biological mechanisms contributing to mitochondrial dysfunction: the uncoupling of oxidative phosphorylation and the inhibition of the electron transport chain. The inhibition of the electron transport chain occurs when chemicals act as alternate electron acceptors, facilitating direct electron transport from complex I to complex IV (or oxygen). These chemicals, exemplified by the 2-nitroaniline moiety, form SA “pairs” where either the oxidized or reduced form can cause mitochondrial toxicity. A similar rationale applies to SAs for hydroxynaphthoquinones/naphthoquinones and hydroquinones/quinones. The uncoupling of oxidative phosphorylation is caused by compounds transporting hydrogen ions across the inner mitochondrial membrane into the mitochondrial matrix. The mechanistic chemistry behind these alerts stems from their role as weak acids in the intermembrane space. In this process, the deprotonated forms scavenge free protons, migrate into the mitochondrial matrix, and undergo proton dissociation in the alkaline environment. This cycle, facilitated by a pH gradient between the intermembrane space (pH ≈ 7) and the mitochondrial matrix (pH ≈ 8), increases oxygen consumption and heat production and decreases the electrochemical gradient and ATP production. The 13 SAs in this category exhibit moieties capable of resonance stabilization, which is crucial for proton cycling.

In a follow-up study two years later, Hemmerich and colleagues compiled a dataset of 5761 compounds from various sources, examining different endpoints related to mitochondrial toxicity [57]. By utilizing fragmentation techniques and alerts proposed by previous studies [58,59], they identified 17 highly specific alerts that could provide a further understanding of the potential mode of action. These alerts were combined with a QSAR predictive model, which was then used to screen the DrugBank dataset, which consists of 2278 approved or withdrawn drugs. A total of 52 compounds were flagged with the identified alerts, and further investigation revealed that there is experimental evidence for 6 of those molecules: amrubicin, tolvaptan, epirubicin, thiothixene, teriflunomide, and ponatinib.

Shifting the focus to another KE, namely, the accumulation of FAs, we have recently developed a SAR model using a reduced set of carboxylic acids. The aim was to identify substructures associated with a steatotic response [60]. We found two relevant substructures presenting a carboxyl moiety and an aliphatic chain comprising a minimum of three carbon atoms. While the presence of the carboxyl group was expected, given its ubiquity in the dataset, the distinguishing factor arose from the aliphatic chain. This characteristic could differentiate steatotic molecules from their non-steatotic counterparts, which typically exhibit a ring in their structure.

In 2021, Jain and colleagues analysed enriched chemical substructures in the steatosis-positive class within their dataset [21]. They identified three significantly enriched scaffolds and utilized the ChemoTyper program [61] to pinpoint enriched structural fragments in the positive class. Of note, eight of the substructural patterns they discovered contained halogens, which is coherent with the fact that halo-alkanes and halo-alkenes can potentially induce toxicity and cancer based on their ability to form a stable carbon-centred radical. As the number of halide groups increases, so does the likelihood of radical formation due to the growing electron-withdrawing capacity from the carbon centre. Reactive intermediates can bind to macromolecules and disrupt lipid metabolism, ultimately leading to steatosis.

Focusing on the prediction of general hepatotoxicity, in 2010, a group of researchers performed extensive literature research to identify SAs associated with DILI [62]. They identified 22 SAs through visual analysis and the consideration of therapeutic endpoints after collecting data on more than 1200 chemicals. The researchers mentioned tetracyclines as an example of steatogenic compounds. Tetracyclines prevent the movement of triglycerides from the liver by inhibiting protein synthesis and causing mitochondrial damage. They have been widely demonstrated in both humans and in vitro models to produce steatosis [63].

A team of researchers employed SARpy in 2016 to automate the extraction of SAs to characterize DILI [17]. They gathered information from several human datasets to compile their dataset comprising 950 compounds labelled either hepatotoxic or non-hepatotoxic. Using the similarity index, the researchers utilized a manual clustering process within the VEGA platform to identify SAs. This process led to the identification of 13 SAs, 11 of which were hepatotoxic. They also utilized SARpy to automatically extract 75 SAs, 40 of which were related to hepatotoxicity. While the authors were able to provide a mechanistic explanation for some of the SAs they identified, they were unfortunately unable to directly relate any of the SAs to steatosis.

In conclusion, SAR analysis facilitates the identification of SAs, and, with the input of experts’ knowledge, it enables the explanation of the underlying biological mechanisms related to the interaction of compounds with specific substructures and the prediction of the potential implication of specific proteins or enzymatic cascades in the occurrence of an AO.

### 2.2. QSAR

Quantitative structure–activity relationships (QSARs) are statistical models that explore the correlation between molecular descriptors encoding the intrinsic nature of the chemicals studied and a determined property or biological activity [64]. These models, often developed through ML algorithms, rely on a clear structural definition of molecules associated with the property or activity of interest [65,66], as happens with a SAR analysis. However, unlike SAR models, which are typically qualitative, QSAR models can provide both qualitative classifications (e.g., toxic vs. non-toxic) and quantitative predictions (e.g., IC50 values, binding affinities, etc.) [67].

The process of constructing a QSAR model follows a systematic workflow comprising several key steps, as illustrated in Figure 4: data gathering and curation to ensure that the available information (chemical structures and property of study) is suitable for modelling; the calculation of molecular descriptors; feature selection to identify the most relevant descriptors; model training using ML algorithms; validation to ensure its predictive performance and reliability; and determination of the model’s AD [47,68]. The quality of a QSAR model depends on factors such as the quality of input data, choice of descriptors, and statistical methods for modelling and validation [69].

In QSAR modelling, several types of molecular descriptors exist, such as physicochemical (e.g., molecular weight, solubility, etc.); graph-theoretical descriptors, which involve counts of atoms and fragments; mathematical descriptors that contribute complexity metrics; fragment-based descriptors that use predefined fragments, chemical groups, or molecular fingerprints; and 3D descriptors that consider volumetric aspects and molecular shape [33,66,70]. These descriptors accurately represent molecular structure or properties and are crucial in establishing a correlation between biological activity or physicochemical properties in QSAR models. In recent years, biology-related fingerprints were introduced as descriptors for QSAR models [71], such as, e.g., compound/pathway interaction fingerprints, as outlined in Füzi et al. [72].

In the last two decades, ML techniques have revolutionized QSAR modelling, enabling more accurate predictions of chemical properties and activities. Traditional QSAR methods have relied on congeneric series of compounds and predefined molecular descriptors, which limited their ability to predict complex biological phenomena [73]. With the advent of ML, QSAR modelling has broadened its scope [64,73]. Techniques such as random forest and Support Vector Machine have become popular for their robust predictive capabilities.

More recently, deep learning, particularly deep neural networks, has shown superior performance in QSAR applications. Deep neural networks can model chemical properties directly from molecular structures, like SMILES strings, without relying on pre-extracted descriptors [67,70]. This data-driven approach departs from traditional ML, offering greater flexibility. Deep neural networks can also perform multitask learning, which is especially beneficial when biological endpoints share a similar mode of action [64,73].

However, deep neural networks face challenges, including high computational demands, interpretability issues, and the need for large datasets. In some cases, simpler models, such as single-layer neural networks, may perform comparably to traditional ML methods, especially when data scarcity or experimental errors limit accuracy [64]. Despite these challenges, ongoing advancements in deep learning are expected to enhance QSAR model accuracy as more sophisticated architectures and larger datasets become available [73].

Ensuring the reliability and predictive power of QSAR models requires robust validation strategies. The first step is the evaluation of the statistical parameters (e.g., coefficient of determination, Matthews correlation coefficient, accuracy, sensitivity, and specificity). Typically, the dataset is partitioned into training and test sets, with approximately 25–35% of molecules allocated to the test set, which is a significant proportion of the total dataset. Model performance can be further assessed using leave-one-out cross-validation and k-fold cross-validation, which offer insights into goodness-of-fit and robustness [69]. On the other hand, external validation evaluates the model’s predictive ability on independent datasets not used during model building, demonstrating its capability to predict outcomes for new compounds [68]. To avoid chance correlations, overtraining, or weak reproducibility, additional strategies, such as using additional independent datasets and the y-randomization test, are implemented [66]. Moreover, the model’s AD must be established based on the training set or the model’s characteristics, as described elsewhere [46,47]. Together, these validation steps play a crucial role in establishing the credibility and applicability of QSAR models in predicting molecular activities.

Once a QSAR model has been developed and validated, it can be utilized to predict the biological activities or properties of a new molecule whose chemical structure is known, and which enters the AD. To have a reliable SAR or QSAR prediction, we must ensure that the molecule is within the AD of the model and that the probability of the prediction is acceptable for those models where the probability can be measured. In the desirable scenario, the prediction unequivocally meets these two criteria.

On the other side, several scenarios can give unreliable predictions:For those molecules that fall outside the AD, the prediction cannot be considered.For those chemicals that are located near the boundaries of the AD, the prediction will be associated with higher uncertainty.For models where the relationship to the AD cannot be unequivocally established—such as those based on physicochemical descriptors, where the reliability of each input is challenging to be determined—the analysis should focus on non-experimental descriptors.For those predictions with a probability between 0.3 and 0.7, the results of the model will be considered unreliable or inaccurate.

In this context, models built from large training sets and broader chemical space will be more capable of generating predictions for a wider variety of chemical structures than smaller, more structurally and mechanistically homogeneous models [74,75,76].

QSAR models are highly efficient and can be easily automated, enabling the evaluation of a large number of molecules in a short amount of time [65,66,77]. These models can predict not only categorical outcomes but also quantitative values, making them highly versatile tools in cheminformatics.

While QSAR models offer high predictive capabilities, they also present inherent limitations. The main challenge lies in mechanistic interpretation due to the non-intuitive nature of molecular descriptors [49,70]. Additionally, the relationship between bioactivity and descriptors can be challenging to comprehend, lacking clear intelligibility.

QSAR models can predict biological activities, enabling them to predict MIEs, KEs, and AOs. Several studies have addressed the QSAR modelling of different nuclear receptors that are identified as MIEs for steatosis [78,79,80,81]. Additionally, in 2018, Gadaleta and colleagues developed a set of nine QSAR models for in vitro high-throughput screening assays related to MIEs in the AOPs leading to liver steatosis: up- and downregulation of PXR, AHR, and LXR, as well as upregulation of PPARα, PPARγ, and NRF2 (nuclear factor (erythroid-derived 2) like 2) [25]. They used data provided by the ToxCast program [82] and employed strategies to address the issue of unbalanced datasets.

On the other hand, several publications have shown the development of QSAR models to predict mitochondrial toxicity as an essential KE in steatosis. For example, Tang and co-authors used Tox21 data [83] to generate discriminant QSAR models for mitochondrial toxicity [84]. The authors found that a consensus model that combines the results of individual models using different types of molecular fingerprints as descriptors can improve the accuracy of the predictions. Substructure analysis revealed that phenol, carboxylic acid, nitro, and arylchloride are informative substructures for mitochondrial toxicity discrimination. Furthermore, researchers at Bayer published a paper on developing a suite of machine-learning models to predict mitochondrial toxicity, leveraging both mechanistic and cell-painting data [85]. They collected mechanistic assay results for membrane potential, respiratory chain, and mitochondrial function from diverse sources. They built individual mechanistic models using continuous and data-driven molecular descriptors [67] and, in parallel, developed models trained on morphological features identified in cell painting assays, independently or in conjunction with the molecular descriptors, and notably, including cell painting profiles demonstrated a marked enhancement in the predictive performance for mitochondrial toxicity.

In a different matter, we previously developed two QSAR models to predict FA and phospholipid accumulation biomarkers, which can aid in differentiating between steatosis and phospholipidosis [60]. To build these models, we utilized data from high-content screening in vitro assays conducted on HepG2 cells exposed to a set of structurally similar carboxylic acids. Our analysis revealed that changes in chain length, flexibility, and molecular weight were important factors linked to FA accumulation.

Additionally, there have been attempts to develop QSAR models to predict hepatic steatosis as the primary endpoint. In 2020, researchers working on the EuroMix project (https://www.euromixproject.eu/, accessed on 20 September 2024) attempted to predict steatosis using QSAR models for nuclear receptors related to steatosis. However, these models did not perform as well as expected. Consequently, they developed a QSAR model to predict steatosis using a dataset of 207 pharmaceuticals and pesticides identified from in vivo assays on rodents [86]. The model’s descriptors included average molecular weight and SlogP, which is a hybrid atomic calculation of the partition coefficient between n-octanol and water.

That same year, another research study focused on developing binary classification models for four mechanisms that lead to drug-induced liver injuries, including steatosis [87]. The study used data on 257 drugs collected from DILIrank [88] and PharmaPendium [89] datasets, as well as 318 drug metabolites obtained from the ADME database [90].

The study found that including drug metabolite sets improved the performance of the models, highlighting the importance of drug metabolism in the prediction of hepatotoxicity. The researchers utilized 2D fingerprints to identify significant structural characteristics more commonly found in the DILI-positive class. They compared the positive and negative sets and the drug and drug metabolite datasets to study the metabolic activation of compounds. They revealed that moieties such as phenyl and hydroxyl attached to aromatic carbons were more frequently observed in positive drug metabolites than in positive drugs. Conversely, carbon–nitrogen bonds were observed to be less frequent in positive drug metabolites than in positive drugs. The different bioactivation and detoxification processes in hepatocytes can cause these differences. However, these structural differences suggest that it is incorrect to assume that metabolites derived from a steatotic drug would necessarily be steatotic themselves. The metabolic transformations in the liver can significantly alter the toxicity profile of the metabolites, meaning that each metabolite must be individually assessed for its hepatotoxic potential.

A year later, Jain and coworkers explored the development of QSAR models for predicting hepatic steatosis using different combinations of chemical descriptors and predictions of transporter interactions and nuclear receptor activation/deactivation [21]. They used an unbalanced database of in vivo studies on rodents with repeated oral exposure, consisting of 120 active and 921 inactive compounds. Consequently, they explored the use of unbalanced learning techniques like bagging with stratified under-sampling and confidence predictors to train their models. While both meta-classifiers proved beneficial, the study underscores the specific utility of confidence predictors, emphasizing their potential relevance to predictive toxicology. Additionally, despite statistically insignificant performance differences, the researchers recommend further inspection of the role of transporters in hepatic steatosis and advocate for their inclusion into newer AOPs.

The performance metrics of the QSAR models discussed in this section, including accuracy, sensitivity, specificity, and correlation coefficient in the case of Al Sharif’s model for PPARγ [78], are provided in Appendix A.

In summary, the versatility and predictive power of QSAR models, coupled with their ability to analyse large datasets efficiently, make QSAR models a valuable computational tool, particularly in the context of AOP data gap filling.

### 2.3. Read-Across

Read-across is founded on the principle that chemicals with comparable molecular structures are likely to share similar chemical, toxicokinetic, and toxicodynamic properties [91,92]. This approach uses information from chemical analogues with known toxicity within the same chemical group or “category” to assess unknown chemicals [93,94].

A chemical category is a group of chemicals that exhibit a similar biological action pattern and comparable toxicity or properties [93,95]. Similarity can be determined by shared functional groups, with common precursors leading to structurally similar metabolites or a consistent pattern in the variation of properties across the group [95,96]. However, it is important to note the inadequacy of relying solely on structural similarities to predict substance properties. Comprehensive scientific justification integrating various properties is recommended to address the potential toxicological impacts of structural variations [97]. This includes multiple lines of evidence, such as chemical structure, molecular properties, ADME (Absorption, Distribution, Metabolism, and Excretion) profiles, and biological assays [92,98]. Enoch and colleagues highlighted that defining chemical similarity based on a clear MIE yields the most accurate prediction [56], emphasizing the importance of a holistic evaluation of biological and physicochemical properties in read-across assessments.

In the context of a read-across assessment, source chemicals are reference points for toxicity estimation within a category, while target chemicals are the compounds being estimated [99]. Read-across can be applied in four ways: first, “one-to-one”, where a single similar chemical is used to estimate the target chemical (Figure 5A); second, “one-to-many”, where a single analogue is used to make estimates for multiple target chemicals (Figure 5B); third, “many-to-one”, where multiple source chemicals are used to estimate the target chemical (Figure 5C); and, finally, “many-to-many”, where multiple source analogues are used to make estimates for multiple target chemicals (Figure 5D) [92,100,101].

There are two ways to conduct a read-across: the analogue and the category approaches. The analogue approach is used when only a few structurally similar substances are available with no apparent pattern in their properties. In this approach, the results of studies conducted on the source substance are directly applied to the target substance for all parameters, often using a worst-case scenario. This approach is generally susceptible to uncertainty [95,97,101]. On the other hand, the category approach is used when several substances share structural similarities and can be grouped based on defined characteristics and differences. Toxicological properties within the group are expected to be similar or exhibit a regular pattern, and predictions for target substance(s) can be made based on this pattern [97]. If no regular pattern is evident for a specific property among category members, a conservative read-across may be made from a category member with relevant information using a worst-case approach [95]. In this context, a worst-case approach ensures that the predicted effects for the target substance are not underestimated, with scientific explanations potentially rooted in kinetic or potency considerations. One notable advantage of this approach is that it enhances confidence in the reliability of results for individual substances within the category. This is due to identifying consistent patterns of effects within the category, which is a more robust and comprehensive evaluation compared to an assessment conducted on a substance-by-substance basis [101].

The workflow for read-across involves seven key steps: defining substance identity, providing comprehensive documentation, selecting the appropriate scenario, developing a hypothesis, gathering supporting evidence, evaluating assessment elements, and concluding the scientific acceptability of the read-across. These steps have been extensively described in the European Chemicals Agency (ECHA) Read-Across Assessment Framework (RAAF) [95]. Several publicly available tools can be used at different stages of the read-across workflow, as illustrated in Table 2. The characteristics and applicability of these tools have been thoroughly analysed and compared in prior works [97,98,99,102].

When a read-across prediction is performed, the following can enhance the confidence of the prediction:Scientific justification that the target chemical belongs to the category of the source chemicals.Experimental data from source chemicals that fall within the predicted values for the target chemical, indicating interpolation rather than extrapolation.Consideration of multiple source chemicals in the prediction process (a many-to-one approach).Integration of the prediction as part of a weight of evidence alongside data from other sources, such as ADME properties or relevant in vitro studies.

This evaluation of a read-across-based prediction is usually performed a posteriori, and, finally, it requires an expert judgment evaluation, which is very difficult to automatize.

In an attempt to assess this confidence, Myatt et al. introduced a scoring method, not only to read-across, to evaluate the reliability of in silico methods in general. This scoring considers the reliability, relevance and coverage of information [110]. Related to that, the ECHA RAAF also provides relevant guidelines to evaluate these aspects [95]. Nevertheless, as far as we know, there is no available tool to evaluate the confidence assessment in an automatic way.

Read-across is a practical alternative to other in silico techniques, such as SAR and QSAR models, especially when the toxicity data are limited. Unlike SAR and QSAR model development, where a large dataset of compounds with known activity is required, read-across allows experts to use prior knowledge to identify suitable source chemicals. For this reason, it is possible to infer the activity of the target chemical even when only a few chemicals have toxicity data. One notable advantage of read-across is that it allows the interpretability of the association between chemical structure and activity, which is a feature often lacking in QSARs. Additionally, contrasting with the use of SAs, it is possible to predict the absence of a property or effect by using read-across from a negative result. This method is considered valid and convincing if the test design is adequate. Moreover, read-across in toxicology is a versatile approach that can model both qualitative and quantitative data, offering simplicity and transparency in its application [93,94,101,111].

Although read-across has proved to be a very useful tool for risk assessment, it also has several limitations that must be recognized. One significant challenge is the limited number of available source chemicals for a target chemical in small datasets. The effectiveness of read-across depends on data availability and quality for similar chemicals, which poses a limitation in cases where information is scarce or of insufficient quality. Additionally, the potential for inaccurate predictions arises when dealing with source chemicals that exhibit conflicting toxicity profiles. Another significant drawback is the reliance on expert knowledge, which introduces subjectivity and variability into the methodology. The current read-across methods rely heavily on the expertise of the individual conducting the study, which can lead to a lack of reproducibility and objectivity. This subjectivity appears in the selection of source chemicals and the interpretation of data, creating uncertainties and hindering the development of standardized practices [93,94,111].

Read-across has been used in several case studies related to hepatic steatosis. In the EU-ToxRisk project context, a microvesicular liver steatosis case study was developed. A group of 19 aliphatic carboxylic acids was evaluated in vitro for their potential to cause MIEs or KEs described in the AOP for liver steatosis [112]. From this case study, a set of publications were made with different read-across approaches. First, the Organization for Economic Co-operation and Development (OECD) published a read-across case study to assess the potential for 2-ethylbutyric acid to cause liver toxicity [92,113]. They used a group of aliphatic carboxylic acids, some known to cause liver steatosis, as source substances for the approach. Despite activating a single MIE (PPARα), in vitro and in silico data contradicted the hypothesis that 2-ethylbutyric acid might cause liver steatosis after 90 days of treatment. This finding aligns with the trend observed in source chemicals, where MIE activation and the potential to induce lipid accumulation decrease with shorter side chain length. Afterwards, the team published a study to enhance the application of chemical read-across in risk assessment by investigating transcriptomic responses of primary human hepatocytes. The focus was on 18 structurally different analogues of valproic acid known to cause hepatic steatosis in rodents [114]. The study revealed that compounds highly structurally related to valproic acid exhibited similar transcriptomic responses regarding gene expression patterns and potency. They highlighted the importance of integrating biological and physicochemical data to enhance read-across approaches. Later, they conducted a case study on 13 branched aliphatic carboxylic acids that were structurally similar [26]. They aimed to investigate the use of human-based NAMs to establish biological similarity. The study found a correlation between side chain length and the biological activity of the carboxylic acids. Longer-chain chemicals showed more significant activity in terms of MIEs and KEs. The study also highlighted the importance of considering late KEs for confirmation, as progression from MIEs to AOs was not guaranteed.

A different study investigated the limitations of using structural similarity alone for read-across in hepatotoxicity research and proposed integrating in vitro omics technologies to improve accuracy [115]. The investigation focused on three structural analogues of p-dialkoxy chlorobenzenes and identified shared biological pathways aligned with observed hepatotoxicity in the source chemical 1,4-dichloro-2,5-dimethoxybenzene. The study found that exposure to these chemicals induced cytochrome P450 (CYP) 4 family genes, linking them to peroxisome proliferators and emphasising the importance of considering metabolic predictions. They identified several biological pathways altered by these chemicals, including mitochondrial dysfunction and FA β-oxidation. The study demonstrated the potential utility of in vitro toxicogenomics for grouping chemicals in read-across.

In summary, while read-across presents some limitations, it is a valuable tool in toxicology and has a high regulatory acceptance.

### 2.4. Omics

“Omics” is a term used to describe the set of technologies that generate high-content data sets with measurements of gene transcripts, DNA methylation, proteins, metabolites, and lipids [116]. These assays produce large, diverse data sets, requiring bioinformatics tools for processing. Advanced computational techniques, including data mining, ML, and statistical methods, are essential for analysing these data and building prediction models [117]. Analysing omics data is essential to identify significant differences between experimental conditions. This analysis is critical for functional enrichment, which helps to identify altered biological pathways after exposure to a compound. Pathway databases such as Gene Ontology [118] or Kyoto Encyclopaedia of Genes and Genomes [119] can be used for this purpose. While processing vast volumes of data presents challenges, database technologies and new statistical algorithms offer solutions [120].

Recently, omics technologies have given birth to a new branch of toxicology known as toxicogenomics [121], which, by integrating omics data analysis and computational toxicology, investigates toxicity pathways and cellular responses to exposure to chemical substances [9,122,123]. Within the AOP framework, this integration facilitates the identification of downstream effects of MIEs and supports KEs by detecting molecular changes [5,9,14,116]. Most importantly, toxicogenomics provides highly valuable mechanistic information and is crucial in constructing AOPs by offering information on KEs, pathway enrichment, and biomarkers for toxicity testing [14,124].

A growing number of omics and toxicology databases useful for AOP development are available in the public domain, reducing the need for further experimentation through data reuse. Resources such as PubChem [125], ChEMBL [126], ToxNet [127], DrugMatrix [128], TG-GATEs [129], Connectivity Map [130], LINCS [131], and the Comparative Toxicogenomics Database [132] can be used to retrieve data from in vitro and in vivo experiments for AOP development [14,116,133]. Additionally, Table 3 shows examples of omics-based databases that provide relevant data for AOP studies.

The following section provides more details about the four main omics and how they can be applied in AOP development.

#### 2.4.1. Genomics

Genomics involves studying an organism’s complete set of DNA, including all its genes and their interactions with each other and the environment. It encompasses technologies like gene sequencing and genotype analysis [153]. Genomics can be used for risk assessment by analysing multiple variants across the entire genome and their impact on susceptibility to compound exposure [123]. The individual’s susceptibility to toxicity from exposure to a compound can be determined at the genome level by considering both pharmacokinetic (distribution, metabolism, and excretion) and pharmacodynamic (response of the tissues to the chemical) aspects [154], although it is not possible to detect effects of the exposure to chemical compounds on the individual, as genomics are not dynamic processes [155].

Genomic studies can be particularly beneficial in cases of idiosyncratic DILI, which involves unexpected adverse reactions in susceptible individuals at safe doses for the general population. Genetic polymorphisms, particularly those related to CYP and conjugating enzymes, are among the risk factors associated with idiosyncratic DILI. These polymorphisms are believed to contribute to variations in drug metabolism, which can lead to the accumulation of the compound or production of toxic metabolites [156,157] or alterations of immune-related genes, such as the human leukocyte antigen gene or genes related to cytokines [158]. For example, the influence of genetic variation of PXR on drug metabolism and clearance has been studied to reveal that it influences the expression of multiple CYPs and plays an important role in glucose homeostasis [159].

In 2011, researchers collected information on a list of genes whose hereditary alteration causes hepatic steatosis [160]. They pointed out that although the errors of metabolism observed in these disorders are observed in a minority of patients, they provide valuable information on the metabolic pathways involved in hepatic steatosis. A more recent study by the Genome-Wide Association used 1483 cases of MASLD and more than 17,000 controls [161]. This study identified genetic signals at four locations (chromosomes 2, 4, 19, and 22), with *PNPLA3* confirmed as a risk factor for the full spectrum of MASLD. Additionally, novel signals such as *PYGO1* were associated with steatosis, implicating Wnt signalling pathways in MASLD pathogenesis. Studies on different patient cohorts have identified *TM6SF2* and *GCKR* as key genes in the progression of MASLD [155]. Although MASLD is not directly caused by drug exposure, identifying these polymorphisms can help identify patients vulnerable to developing steatosis when undergoing pharmacological treatment.

Genomics can also be applied to cross-species extrapolation to help discern the differences in response between animal and human models [123,154]. A 2018 review article focused on AHR and analysed differences among tissues, organs, and species, considering AHR ligands as modulators of this nuclear receptor [162]. Analysing data from 20 studies performed on human, rat, and mouse cell lines, they detected differences in the toxic equivalency factors of several ligands dependent on the species.

#### 2.4.2. Transcriptomics

Transcriptomics involves the global measurement of RNA transcripts in a biological system, reflecting the simultaneous transcription of all genes at one point in time [153]. Different technologies are used for transcriptomics analysis, starting with quantitative polymerase chain reaction (qPCR) arrays, suitable for a few dozen genes, while targeted transcriptomic panels (microarrays) are used for hundreds to thousands of genes. RNA sequencing (RNA-Seq) enables comprehensive transcriptome analysis, whereas targeted RNA-Seq (TempO-seq) balances specificity with broader investigation [9,163]. RNA-Seq has garnered particular interest in the study of liver injury due to its wider dynamic range, making it more adept at detecting genes relevant to hepatotoxicity than microarrays [158]. However, this type of data is also difficult to analyse and interpret and requires a high level of genomic annotation.

For the statistical analysis of transcriptomics data to identify significant differences between experimental conditions, two different approaches can be applied: those analysing individual gene expression (e.g., fold change, rank product, etc.) and those considering the entire distribution (e.g., Bayesian, counting methods, etc.) [133,164]. Subsequently, the data undergo functional analysis to assign biological significance to the results.

The analysis of transcriptomics data helps characterize specific changes in the transcriptome, generating “transcriptomic signatures” that can play a crucial role in developing an AOP. Thus, transcriptomic data provides a solid basis for defining MIEs and KEs. Additionally, they can help identify a set of biomarkers associated with the AO, which can guide toxicity testing and hazard identification [8,165].

Transcriptomic techniques have been used to characterize the downstream effects of the interaction of certain compounds with proteins identified as MIEs of the AOP for hepatic steatosis. For example, microarrays were used to identify upregulated genes when human hepatocytes were treated with rifampicin, which is a well-known PXR agonist [166]. It was observed that the expression of genes such as *CYP3A4*, *CYP2B6*, multidrug resistance gene, *ALAs1* and thyroid hormone-responsive *SPOT14* homolog was increased.

Transcriptomic responses have also been used to identify specific genes and pathways associated with KEs related to hepatic lipid metabolism. For example, a study used transcriptome analysis to investigate three steatogenic compounds (amiodarone, valproic acid, and tetracycline) [63]. Amiodarone and valproic acid upregulate lipid metabolism, while tetracycline downregulates mitochondrial functions and fibrosis. The study suggests that all three compounds may affect PPAR signalling. In 2022, a set of 18 valproic acid analogues was assessed using primary human hepatocytes, which revealed differences in potency and highly similar expression patterns for structurally related compounds [114].

Some authors have taken advantage of the large amount of transcriptomics data available in public databases such as those listed in Table 3. For example, Abedini and colleagues used data from DrugMatrix [128] to perform a differentially expressed gene identification and pathway analysis [167]. The pathways identified through statistical analysis included glycolysis/gluconeogenesis, steroid hormone biosynthesis, retinol metabolism, metabolism of xenobiotics by CYP, PXR/RXR activation, AHR signalling, and xenobiotic metabolism signalling.

AbdulHameed and colleagues used pre-existing transcriptomic data on 18 known steatogenic chemicals to identify genes and pathways associated with drug-induced steatosis [168]. They found that pathways like the retinol metabolism were consistent across species and identified mitochondrial toxicity as a KE for steatosis.

Other studies chose a broader perspective and studied DILI using ML approaches on transcriptomic data. In the book Artificial Intelligence in Drug Design, chapter 18 presents an example model built using data from LINCS L1000 [131] and offers a guide on each step to build the ML model from transcriptomics data [169]. Some years prior, information on 77 compounds from the same database was used to identify the pathways altered by each compound [170]. They incorporated the transcriptomic data into a genome-wide metabolic model of hepatocytes to estimate the overall effects of chemical perturbation on the cell. Using this approach, they could identify, on average, 40 metabolic reactions activated under chemical perturbations.

#### 2.4.3. Proteomics

Proteomics is a multidisciplinary field that studies the functional outcomes encoded by genes by studying dynamic collections of proteins, known as proteomes, in living systems [9,120,153]. Proteomic experiments generally collect data on three properties of proteins in a sample: location, abundance/turnover, and post-translational modifications [9,153,171].

The most common technique for protein detection is mass spectrometry (MS), which has recently evolved to more advanced techniques like Tandem MS and imaging MS [149,172]. Isotope labelling techniques, like isotope-coded affinity tags and stable isotope labelling with amino acids in cell culture, can be applied to facilitate protein detection [117,120,173,174].

Proteomics is a powerful tool in toxicology, and two types of proteomic studies are commonly used. The first type focuses on understanding the molecular responses of a biological system exposed to a toxicant, which can provide knowledge of the cascade of events leading to changes in protein expression and function [153,174,175]. The second type of study is dedicated to identifying the molecular targets of the toxicant itself due to proteomics’ broad scope and in-depth analysis capabilities [175]. These molecular targets could be identified as MIEs in the context of an AOP.

Even though proteomics is a valuable technique for toxicological assessment, it has not been extensively used for studying steatosis [155]. Some proteomic studies have generated data on a few compounds that cause steatosis using in vivo or in vitro models. Tetracycline was demonstrated to affect mitochondrial proteins involved in β-oxidation of FAs, especially acyl-CoA dehydrogenase [176]. Braeuning et al. conducted an in vitro study using proteomic analysis to investigate the hepatotoxicity of methapyrilene in rat hepatocytes [177]. They found that methapyrilene caused significant changes in amino acid and ammonia metabolism, branched-chain amino acid metabolism, urea and tricarboxylic cycle enzymes, and lipid metabolism. On top of that, even at non-toxic concentrations, methapyrilene triggered inflammatory responses.

#### 2.4.4. Metabolomics

Metabolomics involves the comprehensive analysis of small molecule intermediates and end products resulting from various biological processes. Categorized into intracellular, extracellular, microbial, and xenometabolome, they reflect the actions of proteins in biochemical pathways, which are influenced by the genome, transcriptome, proteome, environmental factors, drugs, and underlying diseases [117,153,178,179]. Two complementary analytical platforms are used for metabolomics: nuclear magnetic resonance (NMR), standing out for its robustness and non-destructibility, and MS, which provides very high sensitivity [153,178,179,180,181].

Metabolomics can play a role in developing AOPs by contributing to the understanding of toxic agents’ mechanisms through a subfield known as toxicometabolomics [178]. This approach involves studying metabolites as downstream elements of toxicological mechanisms, providing structural and quantitative data to identify new biomarkers and target organs [9]. Toxicometabolomics not only offers insights into the mechanisms underlying KEs but also directly links specific metabolites to KEs within an AOP, thereby enhancing the causal understanding of the pathway components leading to AOs [165,182].

Metabolomics techniques have effectively characterized steatosis in various studies. For instance, a study employed MS to characterize steatosis induction in HepaRG cells through exposure to sodium valproate at four concentrations [183]. The research team discerned distinct patterns between the exposed and control cells, selecting up to 200 features via multivariate analyses that serve as potential biomarkers for steatosis. Among the identified biomarkers were diacylglycerol, triglyceride accumulation, and carnitine deficiency. Initial toxic responses revealed heightened levels of S-adenosylmethionine and mono-acetylspermidine, coupled with a moderate increase in triglycerides. Moreover, the study uncovered novel specific toxicity markers, including spermidines, creatine, and acetylcholine.

A study used a different approach to develop a technique for early detection of drug-induced steatosis [184]. This group opted to identify urinary metabolite changes associated with tetracycline-induced hepatic steatosis in rats at different disease stages. Combining NMR and MS, they revealed that exposure to 125 mg/kg of this compound potentially induced steatosis, as confirmed by histological and triglyceride content analyses. The integrated analysis of these metabolomic platforms, alongside multivariate statistical methods, which identified six urinary metabolite changes across six metabolic pathways. Notably, lysine concentration correlated with liver damage, providing insights into the pathogenesis of steatosis development. The combination of metabolomics techniques demonstrated their suitability for studying the early stages of steatosis before the presentation of morphological alterations.

Focusing on general hepatotoxicity, a series of untargeted metabolomics studies investigated the impact of hepatotoxic and non-hepatotoxic substances on HepG2 cells. A first work revealed significant metabolic changes triggered by toxic insults and facilitated the construction of a discriminant model capable of distinguishing between non-toxic and hepatotoxic compounds [185]. Furthermore, distinct metabolite alterations could be related to oxidative stress, steatosis, and phospholipidosis, unravelling changes in underlying biochemical pathways [186]. A third study was focused on discerning potentially hepatotoxic drugs, quantifying associated alterations and unravelling the underlying toxicity mechanisms, including steatosis [187]. They trained their models with 25 hepatotoxic and 4 non-hepatotoxic compounds to predict toxicity mechanisms. They then tested 87 chemicals to create a “toxicity index” and identified unique pathways associated with steatosis, including the nicotinate and nicotinamide pathways. Correlation between NAD decrease, a key feature of this pathway, and the onset of steatosis was also highlighted.

#### 2.4.5. Multiomics

Multiomics refers to integrating various omics datasets to comprehensively analyse biological systems [188]. In toxicology, analysing single-type data is deemed insufficient, as this provides only a partial view of the biological system [117,188]. Hence, the integration of multiomics data at a higher organizational level is encouraged to enhance a comprehensive mechanistic understanding of toxicological responses in the context of an interconnected network that leads to determined phenotypic characteristics [117,179,189,190].

Recent advancements in computational models enable the integrative analysis of multi-omics datasets for a more holistic approach in biological research. Integrative analysis employs three main methods to analyse diverse data types: concatenation-based integration, transformation-based integration, and model-based integration [117,188,191,192], which are summarized in Table 4.

Multiomics data are complex and heterogeneous, making it difficult to analyse using ML approaches [191,193]. ML models require larger sample sizes for a correct fitting and can suffer from overfitting challenges if dimensionality problems arise from having too many omics variables associated with insufficient samples [117,188,191].

Another challenge is missing data, which can be omics-wise or sample-wise [117,188,191]. Omics-wise missing values occur when some samples have data for only certain omics, while sample-wise missing values can occur randomly or due to technical and experimental limitations [188,191]. To deal with this, the datasets must be harmonized, and missing values can be imputed; however, this can introduce bias [117,188].

Beyond the technical challenges of integrating diverse omics samples, which leads to the discussion on how distinct omics layers should be assessed at different temporal points [194], temporal data pose challenges for ML algorithms, which assume static inputs [117]. Addressing these issues is crucial for accurately interpreting time-dependent changes in omics data and optimizing the application of ML algorithms in dynamic contexts.

Some studies have performed experiments on multiple omics in the context of steatosis. For example, Ruepp and colleagues performed transcriptomic and proteomic analyses of acetaminophen toxicity in mouse liver [195]. The study revealed acute induction of *GM-CSF* mRNA (*GM-CSF* is a granulocyte-specific gene) from Kupffer cells and decreased mitochondrial chaperone proteins (Hsp10 and Hsp60) before observable morphological changes.

Other multiomics studies, particularly dual-omics studies, take a more comprehensive approach by conducting integrative analyses, delving into pathway analysis to uncover mechanistic insights associated with a steatotic response.

In 2018, van Breda and colleagues conducted an in-depth exploration of the impact of repetitive valproic acid treatment on primary human hepatocytes, aiming to elucidate the mechanisms leading to steatosis [196]. Employing microarrays and integrated data analysis, the team analysed whole-genome gene expression, DNA methylation, and miRNA changes in primary human hepatocytes exposed to a non-cytotoxic dose of the drug for five consecutive days, with a subsequent 3-day untreated period to assess the persistence of omics changes. The study identified significant alterations in pathways related to FA metabolism, cell signalling, bile acid metabolism, nuclear receptor pathways, and biotransformation. Integrative data analyses, including a washout period, revealed persistent changes, particularly on the epigenome level. Surprisingly, valproic acid treatment inhibited key transcription factors (*HNF1A* and *ONECUT1*), disrupting FA and glucose metabolism. The study also uncovered novel genes and networks associated with exposure to this compound and steatosis in a human liver model, shedding light on the molecular mechanisms involved. Notably, the research highlighted the substantial role of valproic acid-induced epigenetic changes, particularly in nuclear receptors such as PPARα, PPARγ, AHR, and CD36, providing new insights into the intricate molecular pathways underlying -steatosis induced by this drug.

The same workgroup published a second multiomics investigation to understand the intricate molecular events associated with mitochondrial dysfunction induced by the same compound in primary human hepatocytes [197]. They employed epigenomic, transcriptomic, and proteomic data to evaluate the cellular response over time. This study was focused on elucidating alterations in pathways related to mitochondrial function, particularly mitochondrial complex I and III–V genes, citric acid cycle, and β-oxidation. Employing time-resolved supervised data integration, the team utilized a network-based approach to reveal dynamic gene interactions during valproic acid treatment. Surprisingly, the study proposed a novel mitochondria–nucleus signalling axis (*MT-CO2–FN1–MYC–CPT1*) and challenged the conventional notion that oxidative stress solely triggers the mitochondria–nucleus signalling pathway. The results unveiled transient methylation changes in mitochondrial DNA and highlighted *MYC* and *FN1* as pivotal players in the persistent mitochondrial dysfunction caused by this compound. They also emphasized the importance of temporal resolution in unravelling complex biological responses.

#### 2.4.6. Omics Strengths and Limitations

Omics technologies, including transcriptomics, proteomics, genomics, and metabolomics, each bring unique strengths and limitations to the development of AOPs. Careful consideration of the research question and the desired depth of understanding is crucial for selecting the most appropriate omics technology in AOP development. Figure 6 shows a visual representation of the strengths and limitations of omics technologies based on their characteristics.

With its high-throughput sequencing techniques, genomics allows cost-efficient sequencing of complete genomes and simultaneous gene expression analysis. However, biological effects from exposure to a toxicant cannot be evaluated solely based on genomic analysis due to post-transcriptional, post-translational, and epigenetic changes [178].

Transcriptomics provides the most comprehensive platform, allowing for the measurement of individual gene expression, isoforms, and post-transcriptional modifications of the complete genome. While this technology provides abundant data, it faces challenges in the robustness of signals and processing of high-dimensional and noisy data [8,124,133].

Proteomics is a technique that analyses thousands of proteins simultaneously, linking proteomic profiles to phenotypes. It is distinct for its focus on post-translational modifications, which are crucial for understanding protein functions and signalling pathways [175]. This focus enhances the precision of pathway analyses and is key for studying protein networks and making quantitative comparisons across biological states [120,198,199]. Despite its advantages, proteomics faces challenges such as being labour-intensive, having a limited dynamic range—which complicates detecting less abundant proteins—and requiring expensive equipment [153,155,199]. Additionally, the proteome’s complexity and instability, along with limited reproducibility, present significant obstacles [178].

Metabolomics is centred on examining the outcome of biological activities, providing a deeper understanding of the biochemical foundations of different phenotypes. It offers advantages such as cost-effectiveness, the simultaneous analysis of a large set of metabolites, and the non-invasive analysis of biofluids. Moreover, shared metabolic pathways and universal metabolite structures enable the translation of metabolic analyses between animal models and humans. However, they also have drawbacks, such as the need for analytical experts and specialized computational skills and the difficulty of extracting and analyzing metabolites with a single analytical platform. Additionally, sample collection, preparation, and analysis can limit metabolite detection in metabolomics research, underscoring the need for careful consideration when employing metabolomics [153,178,180,181,200].

Common challenges within omics technologies collectively impact their robustness and applicability in toxicology studies. Until very recently, the field has faced challenges such as a lack of standardization and regulations for data management, analysis, and interpretation, leading to variability between measurement platforms and bioinformatics pipelines [8,116,201]. Furthermore, omics technologies have intrinsic technical limitations. The data generated typically provide relative concentrations rather than absolute values, introducing uncertainties related to background noise, sample quality, and variability between MS runs and hindering reproducibility [155,201].

The choice of bioinformatics methods influences the outcomes, making data interpretation pipeline-dependent and hindering cross-platform correlation [124]. Furthermore, in certain instances, the annotation of biomolecule functions is constrained, complicating the interpretation of their contributions to the progression of an AO [9].

Despite the challenges, omics data analysis offers a comprehensive insight into an organism’s biology, generating vast amounts of biological data across different complexity levels. This wealth of information becomes a valuable resource within the AOP framework, capable of contributing to virtually all information blocks in AOPs. Recognized for its role as a valuable source of mechanistic information, omics data play a crucial role in unravelling the intricate molecular details that underlie biological responses [8,121,124].

It is important to understand that pre-existing omics datasets cannot be used as predictive computational models. However, the information in these datasets can be utilized to conclude specific omics experiments, and these findings can be applied to novel compounds that belong to the same chemical space. The criteria for evaluating the AD used for read-across and predictive in silico techniques, such as SAR and QSAR, which can be applied to determine whether new compounds belong to the same chemical space.

### 2.5. Structure-Based Approaches

Structure-based approaches are computational techniques that utilize the three-dimensional structures of molecules to study their interactions. In toxicology, these methods can be used to understand the interaction between toxicants and biological systems. Their applications include predicting toxicity outcomes, studying the mechanisms of action of toxicants, screening potential toxins, and facilitating the design of chemicals with enhanced safety profiles. Structure-based molecular modelling techniques primarily encompass molecular docking and molecular dynamics (MD) simulations [65]. These methods allow for the detailed analysis of molecular interactions, binding affinities, and dynamic behaviours of toxicants within biological targets, providing valuable insights into their potential toxic effects.

Molecular docking is defined as the prediction of the conformation and orientation of a ligand within the binding site of a target protein [202]. It is used to understand the modulation of biochemical processes related to a ligand’s activity [203]. The docking process involves the spatial and energy matching between ligand and receptor, allowing the ranking of chemical databases according to complementarity to a given target [204,205]. This in silico method has gained popularity since the early 1980s and consists of two steps: considering the ligand’s conformational state and evaluating its interactions with the target using scoring functions [202,203,204]. Figure 7 illustrates the result of docking a ligand to the LXRβ and shows the specific interactions.

Molecular docking is used in drug discovery for hit identification and optimization, drug repurposing, and multitarget ligand design [66]. Elucidating the interactions between ligands and molecular targets provides insights into disease mechanisms. Advances in docking algorithms and the availability of 3D structures of potential targets, facilitated by the evolution of techniques such as X-ray crystallography and cryo-electron microscopy, have made large-scale screenings possible. These advancements allow for the identification of protein binding sites, novel molecular targets of known ligands, and potential adverse drug reactions [202,204].

The docking process highly depends on the availability of a three-dimensional structure of the target with sufficient resolution [65,202,204]. When such structures are unavailable, in silico homology modelling is often used, which characterizes an unknown protein structure using a related homologous protein whose experimental structure is known [204,206]. Recently, thanks to AlphaFold, DeepMind’s artificial intelligence model [207], structures from the AlphaFold Protein Structure Database can also be used for structure-based drug discovery. However, there are limitations associated with this strategy, such as the absence of a ligand in the structure and the incomplete model that is obtained due to the lack of water molecules, metal ions, and co-factors [206].

According to the “induced-fit” theory created by Koshland [208], both proteins and ligands are inherently flexible and are reshaped by their interactions. Different strategies are used for docking purposes regarding flexibility, going from exclusively rigid docking to semi-flexible and flexible docking. Rigid docking is the fastest but less precise, while flexible docking is the most accurate but computationally expensive. Semi-flexible docking is popular when computer resources are limited [203,205].

On the other hand, for simplicity and faster calculations, the participation of water molecules in protein–ligand interactions is often ignored [202]. Furthermore, when docking is applied alone, the kinetics of possible ligands are not considered [209]. This kind of approximation in the scoring functions is one of the sources of a potential lack of correlation with the experimentally obtained binding affinities [204].

Multiple molecular docking and molecular dynamics tools are available, utilizing distinct algorithms and serving different purposes [65]. Table 5 presents a compilation of several of these programs.

Moreover, beyond the traditional docking approach, scientists leverage other cutting-edge techniques, such as MD. This computer-based methodology emulates the natural movements of atoms and molecules within chemical and biological systems. Using MD simulations, researchers can delve into how chemicals interact with biological systems, such as proteins and cell membranes [227,228]. This, in turn, can offer valuable insights into molecular mechanisms and enable the development of safer chemicals. Given their versatility, MD simulations are extensively employed in computational studies.

A variety of MD simulation software has been developed over time. The most well-known software packages are listed in Table 5. Some MD simulation packages come with their own force fields, while others require external force fields as input for the simulation algorithms [227].

MD simulations can be used to gain insights into the quality of homology-modelled protein structures, investigate the structural or conformational changes of proteins, understand important interactions between a protein and small molecules, and estimate protein–protein and protein–ligand binding affinity [227]. Utilizing MD simulations for virtual screening proves to be a daunting task, primarily due to their high computational demands and extensive time requirements [228], surpassing those of molecular docking.

Docking and other structural approaches are the most useful when the study’s objective is the detection of substrates or inhibitors of determined proteins. Therefore, they can be used to predict changes in the MIEs. Molecular modelling approaches have been employed to study different proteins marked as MIEs in the steatosis AOP network.

For example, LXR was characterized using different molecular modelling approaches and a QSAR model to predict potential binders [79]. The objective of this study was to characterize LXR’s ligand binding domain by defining the essential features leading to binding. For this, the researchers used ensemble docking, which consists of docking a ligand into an ensemble of protein structures corresponding to the same protein. The algorithm automatically selects the ligand–protein pair with the best docking score. In addition, they integrated this approach with identifying key features from a potent binder compound and calculating fingerprints for chemical similarity assessment, comparing with two reference ligands as templates.

Researchers employed a multifaceted approach to explore hepatic steatosis induced by novel replacement flame retardants [229]. In addition to conducting a series of in vitro and omics assays to delineate KEs within the steatosis AOP, they performed molecular docking studies on PXR and PPARγ. The selection of PXR and PPARγ as focal points stemmed from examining the ToxCast database, revealing that most of the novel replacement flame retardants exhibited activity specifically on these two receptors.

Several studies have investigated the mode of action of different compounds as ligands of PPARγ in relation to liver steatosis. In 2014, Tsakovska and colleagues used molecular modelling techniques to analyse PPARγ complexes with full agonists [230]. They characterized the receptor binding pocket and the specific ligand–receptor interactions of the most active ligands. They determined the key residues with which agonists interact and pointed out that the binding pocket presents certain flexibility depending on the bound ligand. Additionally, they developed a model that collected the three-dimensional arrangement of chemical features essential for the interaction with PPARγ from these ligands. In a second study, the same team performed a virtual screening based on docking filtering to screen for PPARγ full agonists [78]. When validating their virtual screening protocol, they obtained an 85% sensitivity, indicating the protocol’s ability to correctly identify PPARγ full agonists. They also achieved a 77% specificity, demonstrating the protocol’s effectiveness in correctly identifying PPARγ decoys (compounds that mimic binder’s properties but differ topologically to reduce the chance of binding).

Still focusing on the same MIE, several studies have investigated the effect of concrete groups of compounds on that receptor. Alemán-González-Duhart and colleagues worked on developing thiazolidinediones derivatives that are effective in treating diabetes by activating PPARγ but would reduce the tendency to cause hepatic steatosis [231]. They screened 130 derivatives and analysed how the differences in their structures changed the interaction with the ligand binding domain of PPARγ. Another study centred on the hepatotoxic effect of bisphenol S used docking to determine the interactions of this compound with PPARγ [232]. They detected interactions with four key residues in the ligand binding domain. The authors concluded that PPARγ is a potential target of bisphenol S, which would explain the steatogenic effect of this compound.

On the other hand, a recent publication used molecular docking in combination with QSAR models to predict mitochondrial dysfunction, which, as mentioned, is considered a KE for steatosis. In this case, the enzyme succinate–cytochrome c reductase activity was taken as a single molecular activity related to mitochondrial dysfunction [233]. This unusual approach can be taken to broaden the applicability field of molecular docking to generate predicted data that could fill knowledge gaps in an AOP network. Another study combined screening with structure-based pharmacophore models with ML models to identify mitochondrial toxicants [234].

An example of automatization of structure-based techniques is the online tool called Endocrine Disruptome (http://endocrinedisruptome.ki.si, accessed on 20 September 2024) that encompasses an array of 14 nuclear receptors, seven of which have been identified as MIEs of the steatosis AOP: ERα, ERβ, GR, LXRα, LXRβ, PPARα, and PPARγ. It is a docking server that utilizes AutoDock Vina for automated molecular docking to validated crystal structures of these nuclear receptors [235].

## 3. Regulatory Acceptance

In the context of risk assessment, several attempts have generated regulatory guidelines to standardize these computational approaches. Organizations like the OECD and the ECHA have worked to generate several documents that delineate principles ensuring the transparency, consistency, and acceptability of results derived from various in silico approaches.

Over the past two decades, legislative measures such as the Registration, Evaluation, Authorization, and Restriction of Chemicals (REACH) regulation have embraced computational approaches like SAR, QSAR, and read-across as valid tools for risk assessment. Under this regulation, manufacturers are obligated to provide detailed information on the chemicals they manufacture, market, or import [77,236,237,238,239].

Comprehensive guidance documents on read-across have been published by the ECHA [100] and the OECD [101]. These documents outline general considerations for conducting read-across [100,101], including the following:Evaluating the relevance and reliability of data from source chemicals.Ensuring that the data align with current OECD test methods.Assessing the impact of multifunctional compounds on read-across reliability.Analysing the purity and impurity profiles of both target and source chemicals to identify impurities that may influence overall toxicity.Comparing physicochemical properties such as molecular weight and water solubility to gauge similarity.Understanding likely toxicokinetics and metabolic pathways.Using QSARs for decision making on additional testing.

In their 2008 guidelines, the ECHA establishes the RAAF under the REACH regulation [100]. Furthermore, the use of NAMs has been explored to support read-across in a consecutive report that gathered several cases of framework application [3].

On the other hand, the OECD has outlined general rules for assessing the suitability of SAR and QSAR models for regulatory purposes. The agreed OECD principles for SAR and QSAR models determine that, to be considered for regulatory purposes, a model should include the following information: a defined endpoint; an unambiguous algorithm; a defined domain of applicability; appropriate measures of goodness-of-fit, robustness and predictivity; and a mechanistic interpretation, if possible [239]. Since 2004, the OECD has published a complete guideline document for validating SAR and QSAR models used for regulatory purposes, where specific steps and requirements are stated [76,239].

The OECD recently developed a guidance document called the OECD Omics Reporting Framework (OORF) to report omics data for regulatory purposes [240]. The framework aims to promote data sharing, increase transparency in data processing, enable the assessment of study quality, and encourage reproducibility for omics toxicology experiments. The OORF resulted from previous efforts to standardize omics technologies in the regulatory field, such as the Transcriptomics Reporting Framework, which was proposed in 2017. The Transcriptomics Reporting Framework provided a detailed list of items that should be reported in a transcriptomics experiment, including experimental design, platforms, and data transformations [124,164]. The new OORF is focused on transcriptomics and metabolomics, leaving work on proteomics for the future. They harmonized modules for the reporting of the study summary and the toxicology experiment, as well as different data analysis methods, and they developed separate reporting modules for data acquisition and processing in transcriptomics assays (microarrays, RNA-Seq or Targeted RNA-Seq, and qPCR arrays) and metabolomics assays (MS and NMR) [240]. The regulation and standardization of omics data can ultimately lead to a more straightforward implementation of analysis workflows to extract any relevant mechanistic information.

In the case of structure-based approaches, although they are widely used for drug discovery purposes and can also be applied in the field of toxicology [78,227,241,242], there have been no efforts thus far to develop regulatory frameworks in the context of risk assessment.

The increasing interest of regulatory agencies in incorporating NAMs, particularly computer-based methodologies, into new-generation risk assessment strategies is expected to drive the establishment of novel regulatory frameworks. These frameworks will standardize the workflows of in silico methodologies, ensuring their reliability for integration into regulatory risk assessment processes.

## 4. Integration of the In Silico Methodologies to Fill Knowledge Gaps in the AOP

During this review, we have examined various computational methods that have been utilized to provide evidence for the activation of different points in the AOP for hepatic steatosis by chemicals. Figure 8 offers a visual representation of AOP points covered by the examples gathered in this review. Our findings are promising, as we could find evidence for many points contemplated in the AOP. We are pleased to report that the majority of MIEs have been covered with SAR, and many have specific QSAR models. We also identified structure-based studies exploring binding to ER, GR, LXR, PPARs, and PXR. We also encountered several examples of SAR, QSAR, read-across, and omics approaches that tackle steatosis as a whole endpoint. However, we observed that only a few KEs identified in the AOP have been addressed in the reviewed examples.

When searching for evidence for each point, we prioritized the key events KEs that are not unique to one of the references AOPs. For this reason, although we are aware of numerous QSAR models for predicting activation and inhibition of CYP450 isoforms [243,244], we did not consider them for this review.

We have found more evidence for the KE of mitochondrial disruption, along with mitochondrial β-oxidation inhibition and dysfunction. This endpoint has not been exclusively studied in the context of hepatic steatosis, as it is an essential indicator of different types of toxicity resulting from exposure to drugs.

In our analysis of existing literature, we utilized the combined AOP as a framework to identify instances of computational approaches employed for predicting specific endpoints. However, it came to our attention that several KEs within the AOP were not addressed in the studies examined. While the upregulation of several genes such as *ChREBP*, *SREBP-1c*, *FAS*, and *SCD1* is highlighted in the AOP and has been extensively characterized, we did not discover any examples in our analysis demonstrating a clear association between the upregulation of these genes and drug exposure. We believe further exploration of omics studies in this area would yield valuable insights into the mechanisms underlying the alteration of these genes. However, this was beyond the scope of our review, which aimed to provide a comprehensive overview of the applicability of computational approaches in addressing data gaps in an AOP.

Moreover, we observed that the elevation in FA influx was marked as a KE. However, we could not identify any computational study that examined the effect of exposure to compounds that directly impact FA influx. While several studies explored changes in lipid metabolism, we do not consider FA influx synonymous with FA metabolism *per se*. Therefore, we conclude that these studies did not address changes in the transportation of FA.

We have observed that while many changes at the cellular level can be identified through in vitro studies (such as cytoplasm displacement or nucleus distortion) these KEs are less commonly used for computational research than mitochondrial disruption. These changes occur because of the accumulation of lipid droplets in the cytoplasm, which displace organelles and compress the nucleus. However, measuring lipid accumulation seems to be the most convenient way to evaluate changes in cell phenotype, and this information has been used as an endpoint for some computational studies to predict cellular changes associated with steatosis.

The question arises of whether new computational models are necessary to predict the specific KEs not covered by currently available models. Firstly, we must consider whether enough data are available for those specific endpoints to build a model. Additionally, we should assess whether having specific computational models for those endpoints provides valuable information. We aim to use all sources of evidence to obtain a more accurate prediction of the steatotic AO. Perhaps focusing our efforts on improving the prediction of the activation of a few KEs and having more specific models for groups of chemicals could be a more practical approach.

On the other hand, we have found some significant changes at the molecular and cellular levels that are not covered in the AOP. For instance, Ivanov and colleagues’ SAR study identified some proteins associated with lipid disorders, but only AHR is mentioned in the steatosis AOP [53]. However, some other proteins they identified, such as the Niemann–Pick C1 protein, crucial in transporting cholesterol and other lipids within cells, are not included. Additionally, other listed proteins have roles in metabolic processes, such as the 1-acylglycerol-3-phosphate O-acyltransferase β in CoA production, fructose-1,6-bisphosphatase in gluconeogenesis, and glucagon-like peptide 1 receptor in the regulation of insulin.

In this review, we have covered multiple omics approaches that offer significant insight into pathways that may not be represented in the AOP network we worked with. For instance, the genes upregulated by exposure to rifampicin in Moreau’s transcriptomics study were absent in the AOP [166]. The proteomics studies we gathered in this work identified several potential biomarkers of oxidative stress that are not currently included in the AOP and could help in the precise characterization of molecular-level alterations. Among the metabolomics studies we reviewed, only Martínez-Sena’s study identified NAD depletion [187]. In contrast, the others identified alterations of other relevant metabolites related to lipid metabolism, such as diacylglycerol and carnitine, which are not reflected in the AOP. These examples illustrate the importance of omics approaches in developing AOPs, particularly in identifying biomarkers that could play a key role as MIE or KE for a specific AO and in aiding in connecting the points in the AOP through integrating molecular pathways.

However, some of the assays used to generate data for in silico modelling discussed in this review present a significant challenge: extrapolating in vivo toxicity from in vitro data. On one side, there is a lack of cellular models to cover all the MIEs and KEs critical for triggering a specific AO. On the other side, in vitro models do not reflect the complexity of complex physiological systems, such as the input from other signalling pathways driven by hormones or cytokines that may influence the outcome. This is especially important if toxicity is caused by metabolites, as some in vitro assays ignore metabolic activation. In the context of the analysis of omics data, there is an additional limitation, as the transcriptome, proteome and metabolome shift in the majority of cells under certain culture conditions, which is evident when different batches are compared [245,246]. Thus, the transparency of protocols and the rigorous and robust reproducibility analysis are essential to understand the limitations and interpret the conclusions.

Building upon the insights gained from the reviewed examples, structure-based approaches, especially molecular docking, provide a comprehensive understanding of protein targets that require precise interaction modes for ligand activation or inhibition. However, in the case of protein targets with promiscuous binding pockets, such as PPARα, more complex methodologies may be necessary to determine interaction specificity. This can include analysing protein–ligand interactions in-depth or performing molecular dynamics simulations. Regardless of the complexity, structure-based techniques can be combined with other in silico methodologies to accurately predict protein activation or inhibition for MIEs.

Based on the information we have gathered, it is apparent that several in silico methodologies, such as SAR, QSAR, read-across, and structure-based approaches, can be used to generate predictions of how novel compounds act on different endpoints in the AOP network. It is important to note that the knowledge generated from omics techniques can be used to develop in silico models to predict the activity of new compounds on those specific endpoints and extrapolate conclusions on novel compounds using read-across approaches. However, linking simplistic in silico models with complex regulatory-relevant in vivo health outcomes remains a major challenge [2]. When multiple sources of evidence are integrated, they must be assessed for their reliability, relevance, and consistency. The weight of evidence assigned to each source should be determined based on these factors. After that, the sources of evidence should be organized and filtered accordingly [99,247,248].

When multiple lines of evidence are available for a single endpoint in an AOP, such as the inhibition of PPARα, it is essential to assess the biological significance of the available predictive models [247]. This evaluation should also consider the biological AD of the MIE within the AOP, including its relevance to taxonomic, age, and sex factors [249]. Additionally, we should assess whether the results obtained for the same chemical using different available models are compatible, thereby assigning a consistency evaluation to the predictions [247]. Finally, we should also evaluate the AD, accuracy and precision of individual computational models used to make predictions for the specific MIE to assess the reliability of predictions made for new chemicals [247,248].

When we encounter inconsistent results for a particular substance using various prediction techniques, we must carefully evaluate all the evidence by examining the specific criteria used to classify the effect [248]. However, establishing a hierarchy of prioritization of techniques can be challenging. Some authors believe that performing a read-across to extrapolate predictions from experimental values of similar compounds can yield more reliable results than those obtained from QSAR models [248]. Nonetheless, the conflicting concept of similarity evaluation discussed in the read-across section can make the criteria for extrapolation unclear, especially when data is scarce.

Conversely, some frameworks have been developed to establish a hierarchy of predictions based on reliability, such as the International Council for Harmonisation of Technical Requirements for Pharmaceuticals for Human Use (ICH) M7 guideline for mutagenicity [250,251]. In these cases, even when different predictions come from the same technique, the final decision is made by experts. Therefore, the best strategy to follow remains an ongoing discussion.

Nonetheless, using computational models presents itself as a promising approach to predict the AO more accurately. By employing this strategy, we can obtain a complete matrix of data that incorporates all the biological knowledge present in the AOP. This approach may be considered more favourable than experimental data, which necessitate the conduction of in vitro or in vivo assays, as well as the associated time and economic costs and ethical implications in the case of in vivo assays.

## 5. Conclusions

In this comprehensive review, we have compiled the characteristics and potential applications of various computational methodologies, including SAR, QSAR, read-across, omics data analysis, and structure-based approaches, in addressing data gaps within an AOP framework or constructing new AOPs. Utilizing a combination of four established AOP networks for drug-induced hepatic steatosis as our focal point, we have critically analysed existing studies, elucidating the strengths and limitations of each explored technique.

The case study on hepatic steatosis demonstrates the versatility of computational techniques in generating a more complete and interconnected data matrix. Our analysis reveals that computational models can effectively complement AOPs by filling critical data gaps and identifying new KEs. The examples gathered in this paper highlight the transformative impact of in silico techniques in toxicology, providing a blueprint for integrating these strategies into new-generation risk assessment methodologies. By leveraging the strengths of computational tools, we can achieve more efficient, cost-effective, and predictive toxicological assessments, ultimately enhancing our ability to safeguard public health and the environment.

There is a growing acceptance by regulatory agencies like OECD and ECHA of these computational methodologies, which underscores their practical applicability in toxicological risk assessments. However, full implementation into regulatory decision making remains a challenge, requiring standardized workflows and consistent validation protocols for computational models. Regulatory guidelines like the OECD principles for QSAR model validation provide a strong foundation but need further refinement to accommodate emerging methods. To achieve that, continuous collaboration between regulatory bodies, industry, and academic research is essential. In conclusion, while hepatic steatosis serves as a pertinent case study, the principles and guidelines proposed in this review are broadly applicable to other types of chemical-induced toxicity. The continued development and integration of computational approaches within the AOP framework is necessary to address existing data deficiencies and to support the establishment of novel AOPs. This integrated approach holds great promise for the future of toxicological risk assessment, providing a more mechanistically driven and evidence-based foundation for regulatory decision making.

## Figures and Tables

**Figure 1 ijms-25-11154-f001:**
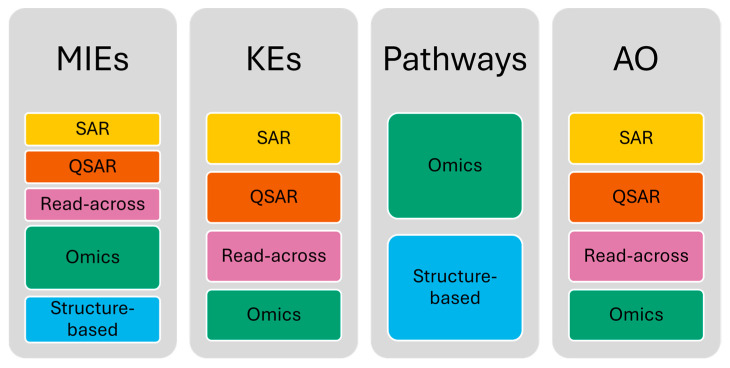
Representation of potential use of computational methods in AOP elements.

**Figure 2 ijms-25-11154-f002:**
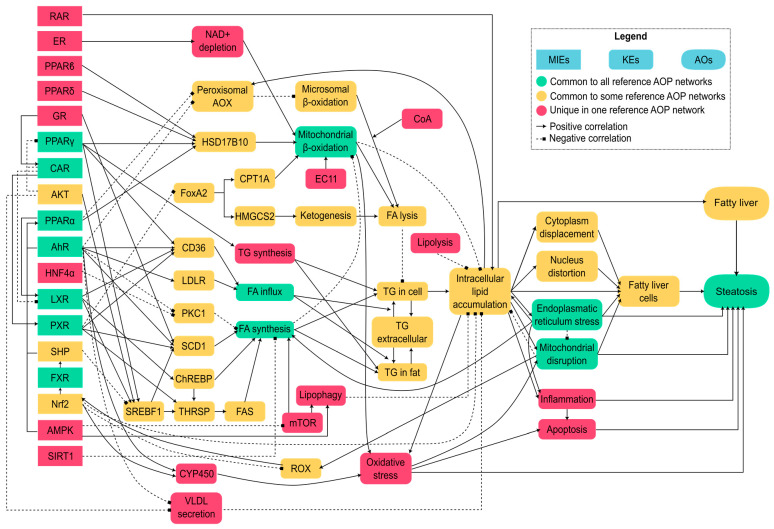
AOP network for liver steatosis. It integrates MIEs and KE collected from four reference AOP networks [11,24,25,26]. The network depicts the progression from MIEs through KEs to the AO of liver steatosis. The nodes are color-coded to indicate their presence in the reference AOP networks: green for nodes common to all reference AOP networks, yellow for nodes common to some reference AOP networks, and red for nodes unique to one reference AOP network. Solid arrows represent positive correlations, while dashed arrows represent negative correlations. The names and abbreviations used in this figure are available in the Appendix A.

**Figure 4 ijms-25-11154-f004:**
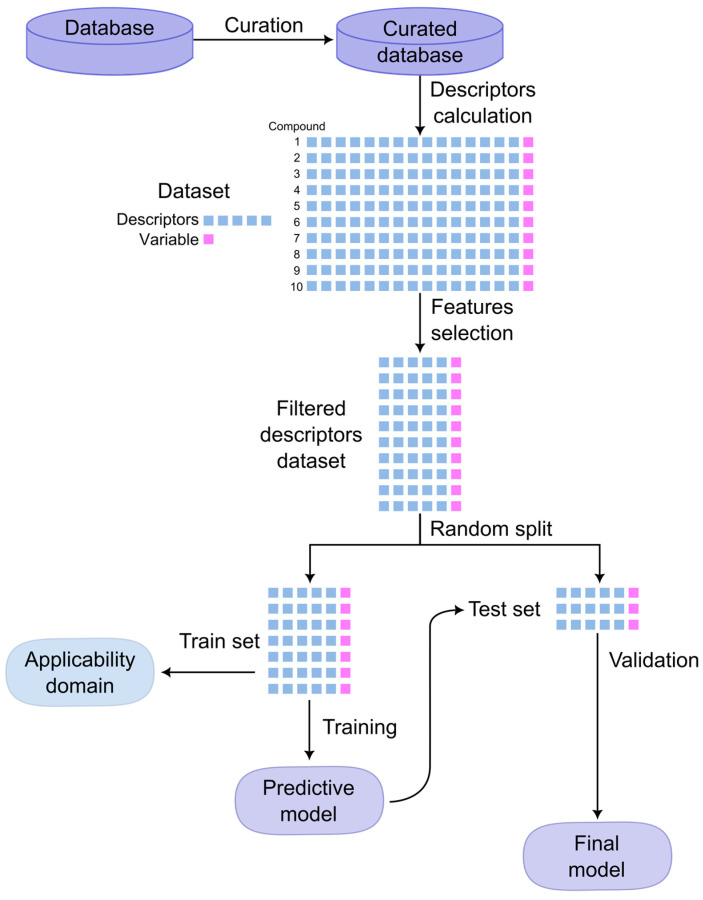
QSAR model generation workflow.

**Figure 5 ijms-25-11154-f005:**
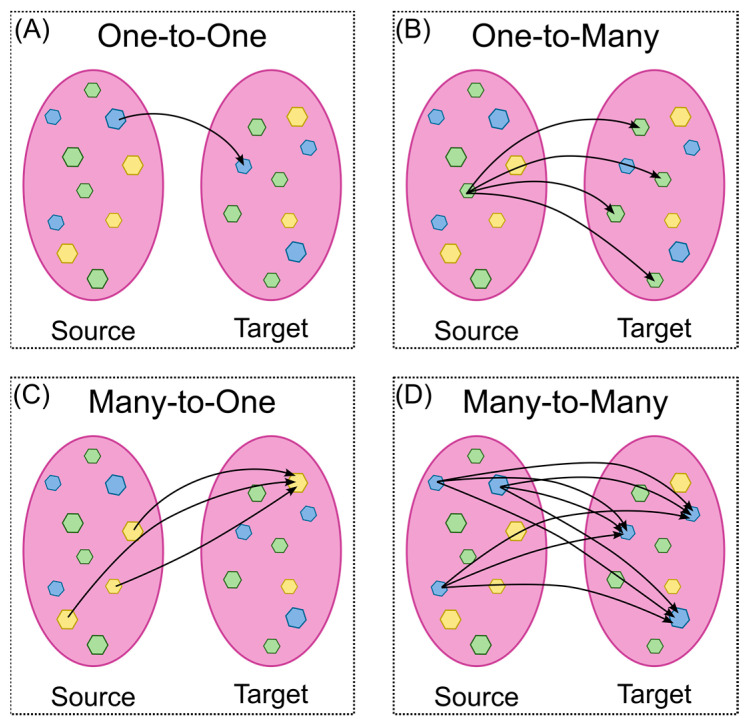
Illustration of read-across assessment approaches. (**A**) One-to-one. (**B**) One-to-many. (**C**) Many-to-one. (**D**) Many-to-many.

**Figure 6 ijms-25-11154-f006:**
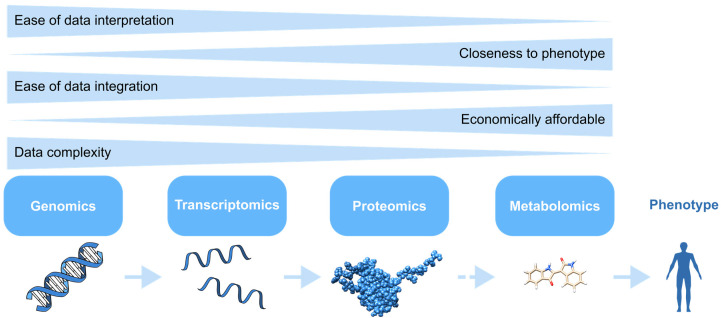
Visual comparison of four omics technologies: genomics, transcriptomics, proteomics, and metabolomics.

**Figure 7 ijms-25-11154-f007:**
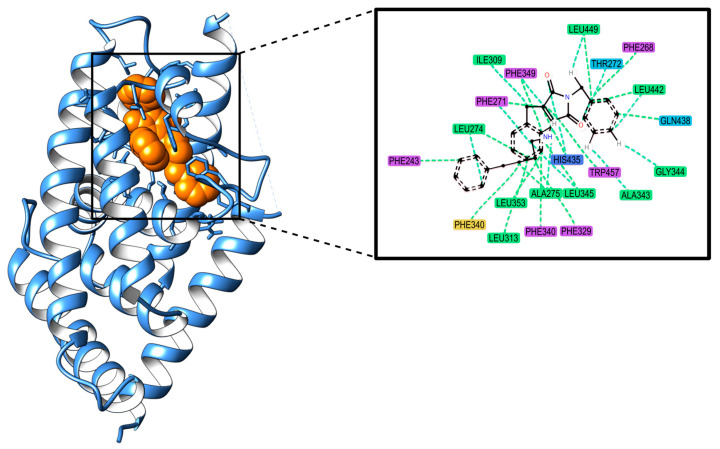
LXR ligand docked onto LXRβ structure. The figure on the right shows a representation of the ligand–protein interactions.

**Figure 8 ijms-25-11154-f008:**
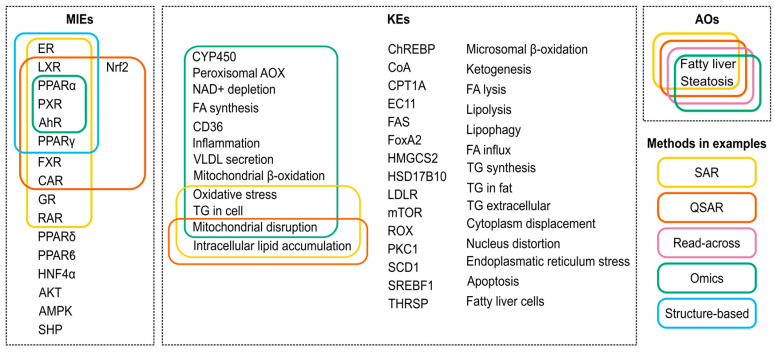
Hepatic steatosis AOP points covered by the different computational approaches reviewed in this work.

**Table 1 ijms-25-11154-t001:** Programs for automatic SA generation. All links were accessed on 20 September 2024.

Approach	Program	URL	Reference
Fingerprints	Bioalerts	https://github.com/isidroc/bioalerts	[38]
Fragments	CASE (Computer-Automated Structure Evaluation), MultiCASE	https://multicase.com/case-ultra	[39,40]
	PASS (Prediction of Activity Spectra for Substances)	https://genexplain.com/pass/	[41]
	SARpy	https://sarpy.sourceforge.net/readme.html	[42]
Graphs	Gaston		[43]
	MoSS		[44]

**Table 2 ijms-25-11154-t002:** Publicly available tools for the read-across workflow. All links were accessed on 20 September 2024.

Tool	Remarks	Link	Reference
AMBIT	Used for substance identification and similarity assessment.	https://cefic-lri.org/toolbox/ambit/	[103]
QSAR Toolbox	Utilized to develop hypotheses and evaluate assessment elements by providing mechanistic and endpoint-specific data.	https://qsartoolbox.org/	[104]
ToxRead	Provides toxicity predictions and read-across justification based on structural and mechanistic similarities.	https://www.vegahub.eu/download/toxread-download/	[105]
ToxMatch	Useful for identifying substances and scenarios by finding similar compounds and creating matrices.	https://toxmatch.sourceforge.net/	[106]
GenRA	Supports hypothesis development and evaluation by offering a read-across methodology that predicts toxicity based on chemical similarity and existing data.	https://comptox.epa.gov/genra/	[107]
ToxGPS	Used for collecting evidence and evaluating assessments by combining data sources to offer detailed toxicity profiles and predictions.	https://mn-am.com/products/chemtunestoxgps/	[108]
VERA	Uses multiple similarity metrics and clusters of analogues	https://www.vegahub.eu/portfolio-item/vera/	[109]

**Table 3 ijms-25-11154-t003:** Omics data repositories. All links were accessed on 20 September 2024.

Omics	Data Repositories	Website	Reference
Genomics	Cell Collective	https://cellcollective.org/#	[134]
	EGA (European Genome-phenome Archive)	https://ega-archive.org/	[135]
	ENA (European Nucleotide Archive)	https://www.ebi.ac.uk/ena/browser/home	[136]
	EVA (European Variation Archive)	https://www.ebi.ac.uk/eva/	[137]
Transcriptomics	ArrayExpress	https://www.ebi.ac.uk/biostudies/arrayexpress	[138]
	dbGaP (database of Genotypes and Phenotypes)	https://www.ncbi.nlm.nih.gov/gap/	[139]
	ExpressionAtlas	https://www.ebi.ac.uk/gxa/home	[140]
	GEO (Gene Expression Omnibus)	https://www.ncbi.nlm.nih.gov/geo/	[141,142]
	SRA (Sequence Read Archive)	https://www.ncbi.nlm.nih.gov/sra	[143]
Proteomics	JPOST (Japan Proteome Standard) Repository	https://jpostdb.org/	[144]
	MassIVE (Mass Spectrometry Interactive Virtual Environment)	https://massive.ucsd.edu/ProteoSAFe/static/massive.jsp	[145]
	PaxDb (Protein Abundance Database)	https://pax-db.org/	[146]
	PeptideAtlas	https://peptideatlas.org/	
	PRIDE	https://www.ebi.ac.uk/pride/archive/	[147]
	ProteomeXchange	https://www.proteomexchange.org/	[148]
	ProteomicsDB	https://www.proteomicsdb.org/	[149]
Metabolomics	HMDB (Human Metabolome Database)	https://hmdb.ca/	[150]
	MetaboLights	https://www.ebi.ac.uk/metabolights/	[151]
	Metabolomics Workbench	https://www.metabolomicsworkbench.org/	[152]

**Table 4 ijms-25-11154-t004:** Characteristics of different omics data integration strategies.

Integration Strategy	Description	Processing	Main Drawback
Concatenation-based	Combines different datasets into a large matrix, increasing variables but not samples.	Applies feature selection or dimensionality reduction.	Makes the data matrix complex and noisy.
Transformation-based	Transforms each dataset into a less dimensional and noisy graph or kernel matrix.	Transformation into graph, kernel matrix, or deep learning architectures.	Can lead to information loss and distorted data.
Model-based	Applies separate ML models to each dataset and combines predictions.	Supervised: Bagging or voting to combine models. Unsupervised: Aggregates clustering results based on optimization criteria.	Prevents learning features shared across different omics data, limiting understanding of underlying mechanisms.

**Table 5 ijms-25-11154-t005:** Molecular docking and molecular dynamics most used tools. All links were accessed on 20 September 2024.

Technique	Tool	Website	Reference
Molecular Docking	AutoDock	https://autodock.scripps.edu/	[210]
	CSAlign-Dock	https://galaxy.seoklab.org/cgi-bin/submit.cgi?type=CSALIGN	[211]
	DOCK	https://dock.compbio.ucsf.edu/	[212]
	DockingServer	https://www.dockingserver.com/web	[213]
	GalaxyDock	https://galaxy.seoklab.org/cgi-bin/submit.cgi?type=DOCK	[214]
	Gold	https://www.ccdc.cam.ac.uk/solutions/software/gold/	[215]
	Glide	https://newsite.schrodinger.com/platform/products/glide/	[216]
	HADDOCK	https://wenmr.science.uu.nl/	[217]
	Medusa Dock	https://dokhlab.med.psu.edu/cpi/#/MedusaDock	[218]
	PyDock	https://life.bsc.es/pid/pydock/	[219]
	Rosetta	https://www.rosettacommons.org/software	[220]
	SwissDock	http://www.swissdock.ch/	[221]
Molecular Dynamics	GROMACS	https://www.gromacs.org/	[222]
	AMBER	https://ambermd.org/	[223]
	NAMD	http://www.ks.uiuc.edu/Research/namd/	[224]
	CHARMM	https://www.charmm.org/	[225]
	Desmond	https://www.schrodinger.com/platform/products/desmond/	[226]

## Data Availability

No new data were created or analysed in this study. Data sharing is not applicable to this article.

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
