# Peer review of "Computational Strategies for Assessing Adverse Outcome Pathways: Hepatic Steatosis as a Case Study"

_ijms, 2024, doi:10.3390/ijms252011154_

Round 1

Reviewer 1 Report

Comments and Suggestions for Authors

Review (IJMS-3243909):

The current review article by Vallabona et al. highlights the transformative impact of in silico techniques in toxicology. Using hepatic steatosis as a case study, the authors explore the use of various computational approaches, such as structure-activity relationship (SAR) models, quantitative structure-activity relationship (QSAR) models, read-across methods, omics data analysis, and structure-based approaches, to fill data gaps within adverse outcome pathway networks. Overall, the review is well presented and discussed. I appreciate the authors’ efforts in providing such detailed views on the computational strategies for assessing adverse outcomes pathways. However, it seems that the omics sections omics sections are somewhat general compared to the disease specifications. Additionally, I observed some formatting and typographical errors. Below are my comments:

Other Comments:

  1. The figure numbers are a bit inconsistent; they need to be arranged and referenced appropriately. For example, Figure 3, should be Figure 4 as Figure 2 is numbered twice
  2. Could the authors mention the names of the compounds in the figure captions (i.e., actual Figure 3).
  3. Define the acronyms upon their first occurrence.
  4. Try to use a consistent format, i.e., either italics or normal font for terms like in vitro, in vivo, in silico, and de novo representation.
  5. Actual Figure 7: Some of the residue labels are hard to read. Please consider changing the residue labels.
Comments on the Quality of English Language

The English language is fine. 

Reviewer 2 Report

Comments and Suggestions for Authors

The article is well organized and provides background information on Adverse Outcome Pathways and tiered chemical risk assessments, followed by a detailed overview of computational methods such as SAR, QSAR, Read-Across and Omics. The article should be of interest to any reader interested in drug design. The inclusion of various techniques is thorough, with the integration of omics data adding depth to the discussion. The explanations of molecular descriptors, validations and limitations of SAR/QSAR models demonstrate sound expertise, while illustrations such as the AOP network help to visualize complex metabolic pathways.

Possible improvements are merely suggestions that you do not have to implement:

- In section 2.2 on QSAR models, the authors should include specific metrics such as accuracy, sensitivity and specificity from real case studies that show the practical performance of QSAR models.

- In the section on SAR/QSAR models, the authors should provide clearer examples of chemical classes or scenarios where predictions may be unreliable and discuss how these limitations affect real-world applications.

- In section 2.3 on read-across, the authors could cite recent advances in confidence assessment tools and examples that demonstrate their role in improving the reliability of read-across predictions.

- A discussion of the limitations of in vitro data and their extrapolation to in vivo predictions, particularly in the context of toxicogenomics and metabolomics, would enrich this manuscript

- In section 2.2 on QSAR models, the authors could also explain the role of machine learning, especially newer techniques such as deep learning, to improve the accuracy and predictive power of QSAR models.

- In the conclusions section, the authors should strengthen the conclusion with a more detailed discussion of the regulatory implications and practical integration of computational methods into toxicological risk assessments.

In my opinion, the work is mature enough for publication. The suggestions are informative, the authors can decide whether to take them into account.

Reviewer 3 Report

Comments and Suggestions for Authors

The authors presented in a well-organized way the review on Computational Strategies for Assessing Adverse Outcomes Pathways for Hepatic Steatosis.

I have no question and this work is really worth for the current research demand. There are few minor points which must be improved.

1. There is issue in Figure numbers such figure 2 appears two times so re-number all the figure sequentially and accordingly update with the text references for figures.

2. The quality of "figure 2. AOP network for liver steatosis" seems blurred and thus must be 600 dpi. and the same needs to be done for "Figure 5. Visual comparison of four omics technologies: genomics, transcriptomics, proteomics, and metabolomics".

3. Typos mistakes are present in many places and thus please perform proofreading after correcting and before submitting the revised version.

Comments on the Quality of English Language

Not required. 
